# Distilling nanoscale heterogeneity of amorphous silicon using tip-enhanced Raman spectroscopy (TERS) via multiresolution manifold learning

Guang Yang [1✉], Xin Li[1,4✉], Yongqiang Cheng[1], Mingchao Wang[2], Dong Ma [1], Alexei P. Sokolov[1,3], Sergei V. Kalinin [1], Gabriel M. Veith [1] & Jagjit Nanda[1✉]

Accurately identifying the local structural heterogeneity of complex, disordered amorphous materials such as amorphous silicon is crucial for accelerating technology development. However, short-range atomic ordering quantification and nanoscale spatial resolution over a large area on a-Si have remained major challenges and practically unexplored. We resolve phonon vibrational modes of a-Si at a lateral resolution of <60 nm by tip-enhanced Raman spectroscopy. To project the high dimensional TERS imaging to a two-dimensional manifold space and categorize amorphous silicon structure, we developed a multiresolution manifold learning algorithm. It allows for quantifying average Si-Si distortion angle and the strain free energy at nanoscale without a human-specified physical threshold. The multiresolution feature of the multiresolution manifold learning allows for distilling local defects of ultra-low abundance (< 0.3%), presenting a new Raman mode at finer resolution grids. This work promises a general paradigm of resolving nanoscale structural heterogeneity and updating domain knowledge for highly disordered materials.

[1] Oak Ridge National Laboratory, Oak Ridge, TN 37831, USA. [2] Department of Materials Science and Engineering, Monash University, Clayton, VIC 3800, Australia. [3] Department of Chemistry, University of Tennessee, Knoxville, TN 37996, USA. [4] Present address: State Key Laboratory of Green Chemical Engineering and Industrial Catalysis, Sinopec Shanghai Research Institute of Petrochemical Technology, 1658 Pudong Beilu, Shanghai, PR 201208, China. ✉email: yangg@ornl.gov; lix3@ornl.gov; nandaj@ornl.gov

Silicon is central for a gamut of applications including large-scale integrated electronic circuits[1], photonics[2], photovoltaics[3,4], and energy storage units[5–7]. It is well known that the essences of Si-based materials, including optical, electrical[8] properties, and nuclear spin[9], are highly related to their atomic structures. Ever since Russell reported the first observation of the first-order inelastic Raman scattering in a Si single crystal[10], Raman spectroscopy has been intensively used to investigate the Si crystal structure[11], phonon dispersion[12], electronic states[13], local stress and strain[14,15], and thermal properties[16], which are integral to the performances of silicon-based devices. Despite its versatility, the applications of micro-Raman spectroscopy to characterize the submicron-to-nanoscale chemistry and physics are severely restrained by the intrinsic diffraction limit of the visible light (i.e. >200 nm) based on Abbe's law[17]. Tip-enhanced Raman spectroscopy (TERS) provides an apertureless means of mapping Raman scattering at the nanometer scale (~10 nm) in-sample plane[18,19]. TERS is based on strong and local enhancement of the Raman signal by the surface plasmon resonance (SPR) on the metallic tip surface. Additionally, a non-resonant enhanced electromagnetic field (EM-field) occurs at the apex of the elongated metallic nanostructure, termed as the lightning rod effect[20,21]. The combination of the SPR and the lightning rod effect enables the EM-field to concentrate on the metallized tip apex of a scanning probe microscope (SPM), providing a localized "hot-spot" underneath the SPM tip. Consequently, the Raman scattering signal of the sample within the local hot-spot is largely increased, yielding surface chemical information with a nanoscale lateral resolution[22]. Sun et al. first reported the successful TERS study on a Si wafer[23]. With a 50% increase in the TERS intensity with respect to Raman, they were able to construct the TERS mapping of the Si transverse optical (TO) mode (520 cm$^{-1}$) at ~100 nm resolution. Later, upon optimization of the polarization conditions, Sokolov et al. improved the ratio of the near-field Raman intensity and far-field Raman intensity by more than one order of magnitude[24], hence being able to carry out the nano-Raman analysis of the crystal Si at a ~20 nm lateral resolution.

Despite great efforts in deciphering the nanoscale vibrational structures on the Si surface using TERS, the research focus has only been on crystal Si (c-Si) so far. Due to the highly symmetric diamond cubic crystal structure of c-Si, its inelastic Raman shift carries the information reflecting optic phonon energy only at the center (i.e. Γ-point) of the first-order Brillouin zone (BZ)[11,25]. Therefore, the spectral variation on two adjacent sampling points is rather vague on TERS mapping.

Different from its c-Si counterpart, the energy states of amorphous silicon (a-Si) vary in the first order BZ, hence resulting in several convoluted Raman active modes. Due to k-selection rule breaking down for amorphous materials, the TERS spectrum taken at each sampling point roughly corresponds to the phonon density of states (PDOS), thereby capable of carrying local information of the a-Si[11]. However, lacking the long-range atomic ordering and symmetries, to quantify the structure of amorphous materials, such as a-Si remains a long-stand challenge. This is especially true for a large TERS dataset gathered from a-Si over a large scanning area, which contains spectra collected from adjacent sampling points of nanometers away sampling points. Accordingly, it is almost impossible to implement a manual exploration over the structural metrics of the a-Si, let alone mining essential information of a miniature structure embedded in such a large dataset. In addition, complex interactions between the tip-induced EM-field and Raman scattering tend to restrict the use of traditional linear dimension reduction techniques, such as principal component analysis (PCA) (e.g. it becomes difficult to extract physical information from unmixed

components)[18,26]. To date, there are a few studies in which TERS was used for studying the structural heterogeneity of amorphous carbon[27]. The nanoscale structural heterogeneity and local structure of a-Si have not been explored using TERS so far.

We herein illustrate a multiresolution analytical framework based on graph-analytics and an unsupervised manifold learning algorithm to facilitate identifying the nanoscale structure of a-Si thin film. High-dimensional hyperspectral TERS mapping on the a-Si thin film comprises thousands of TERS spectra. The multiresolution manifold learning (MML) algorithm projects the TERS mapping to a low dimensional (i.e. 2D) manifold space, thereby allowing for ease of straightforward data visualizations and structural categorization. Unlike traditional manifold learning methods targeting solely on overlaying the prior-known labels over the manifold points[28], the MML proposed here does not require any prior bias regarding the material structure and instrumental modality[29,30]. Benefiting from this nature, the underlying a-Si structural and physical properties, such as the average Si–Si distortion angle and the strain-free energy can be quantified without a human-specified physical threshold at a lateral spatial resolution of <60 nm. Further, the multiresolution feature of the MML algorithm allows for extracting child clusters at the finer resolution grid, which carries structural characteristics of minor abundance (<0.3% of the sampling points) that would have been hidden in the large dataset. It thus enables us to discover a new Raman mode ascribed to highly disordered $O_x$–Si–$H_y$ vibrations on the a-Si surface that has never been reported before. The identification of such a new vibrational Raman mode was further validated through inelastic neutron scattering and density functional theory (DFT). Although a-Si was used as a model, the integrated workflow proposed is readily available for nanoscale structure-property correlation for other highly disordered materials in general.

## Results and discussion

The strategy of effectively employing TERS hyperspectral imaging and the MML to elucidate the local structure of the a-Si is schematically illustrated in Fig. 1. Briefly, TERS combines the scanning probe microscopy (SPM) and the Raman spectroscopy technique, enabling the spectral acquisition at a nanoscale lateral resolution. This is due to that the surface plasma resonance (SPR) on a noble metal (such as Ag) coated tip apex upon laser illumination leads to a greatly enhanced EM-field or a hot spot in the gap between the tip and sample surface (<5 nm in depth) (Fig. S1)[18]. The Raman scattering cross-section from the analyte in the hot-spot is boosted, contributing to the far-field enhanced Raman scattering (i.e. TERS spectrum). The size of a typical hot-spot on the tip apex is on the scale of 10 nm (see Fig. S1d). Scattered light enhanced in the hot-spot is registered on a distant detector in the far-field. Using the raster step size of the SPM probe smaller than the hot-spot size enables to record the nanoscale chemical heterogeneity, breaking the light diffraction limit of the standard confocal micro-Raman spectroscopy[18,24]. However, the routine TERS mapping composes of innumerable spectra, precluding an easy insight of the underlying structure. This is particularly true for amorphous materials. Therefore, it necessitates a more statistical data processing method capable of implementing the batch process of the large quantity TERS spectra to efficiently obtain the structural features within the scanned area.

To explore relationships among all the high dimensional TERS spectra detailing the a-Si local structures, we first construct the nearest neighbor (NN) graph by calculating pairwise distances. For straightforward exploration and visualization purpose, the low dimensional (2D/3D) manifold layout for the NN graph was estimated by solving a principal probability model (details of

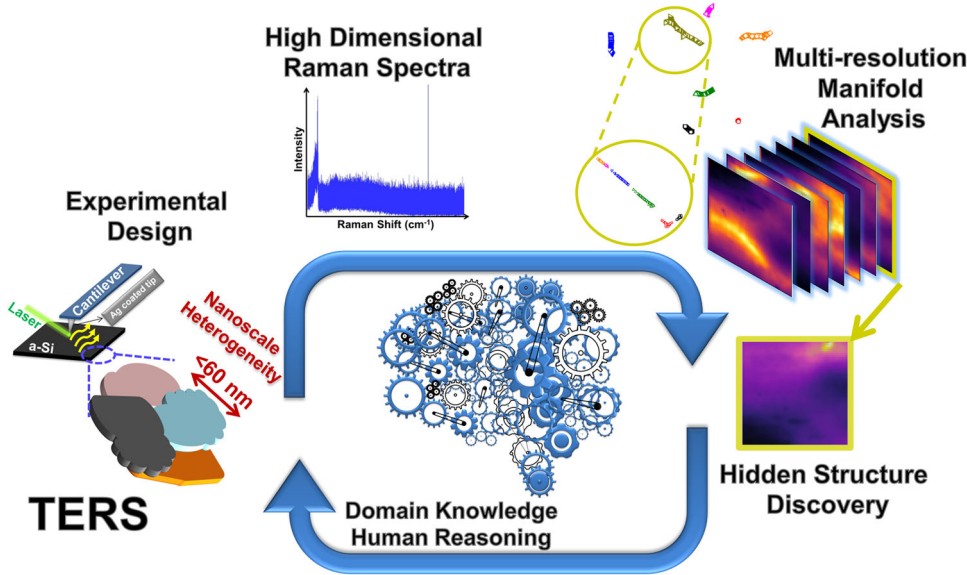

**Fig. 1 The integrated workflow comprising hyperspectral TERS imaging and the multi-resolution manifold learning (MML) algorithm.** The low dimensional physical parameters characterizing material structures such as local defects and atomic vibrations are translated into the high dimensional Raman spectra via hyperspectral tip-enhanced Raman spectroscopy (TERS) imaging transfer functions. The intrinsic low dimensionality of the physics suggests the structure of TERS measurements on the sample as a whole can be projected to a low dimensional latent manifold space via the multiresolution manifold learning (MML). Exploration data analysis such as clustering can be efficiently conducted on the low-dimensional manifold space to reveal salient features for evaluating material structure heterogeneity by human reasoning and updating domain knowledge in a loop.

graph construction and manifold layout can be found in the "Method" section). Clustering can be subsequently performed on the low-dimensional manifold to underpin the intrinsic structure within the manifold that corresponds to the material structure heterogeneity. To better partition intrinsic manifold clusters (facilitating clustering tasks), Li et al. recently proposed a Graph-Bootstrapping procedure[29,30] that iteratively reconstructed the NN graph based on previous manifold positions and then recalculated manifold coordinates based on the reconstructed NN graph. The projected low dimensional manifold clusters represent featured spectral property of the materials, thereby allowing for gaining insights of latent material structures via external validations such as first principle theories. This, in turn, benefits a future experimental design with human-reasoning for gaining a deeper structure–property relationship of the materials (Fig. 1).

The characteristic TERS spectra obtained from the a-Si surface are shown in Fig. 2a. All spectra contain a sharp peak located at $520\,cm^{-1}$, ascribed to the first–order TO mode for the crystal silicon derived from the TERS (Fig. S3). For c-Si, only the zone-center TO mode is detectable based on the excitation of the visible Raman laser to the lattices of the diamond structure, with the Si–Si bond angle of $109.5°$[10,11,31]. Notably, the coexistence of the a-Si and c-Si TO modes in Fig. 2a does not indicate that the a-Si is partially crystallized on the surface[32]. A further inspection on the a-Si surface using confocal micro-Raman spectroscopy with a much higher laser power did not reflect the existence of the c-Si TO band at $520\,cm^{-1}$ (Fig. S3). Instead, a few broad Raman bands were featured in several frequency regions, indicating that the sputtered-silicon in this study is in the totally amorphous state. The broadening of the several Raman bands results from the loss of the long-range translational symmetry and corresponding reciprocal lattice of the c-Si, which allows for detection of the entire phonon density of the states (DOS) across the whole first BZ zone reflected by the Raman spectra[33]. Though it is inappropriate to correlate the traveling wave vectors to the phonon feature in amorphous solids, we assign the a-Si TERS bands to phonon frequencies for ease of comparison with

numerous other studies[34–38], including the modes of the longitudinal-acoustic (LA, $312\,cm^{-1}$), longitudinal-optical (LO, $400\,cm^{-1}$), and transverse-optical (TO, $473\,cm^{-1}$), respectively[34]. The second-order phonon modes (denoted as 2LA and 2TO) of either a-Si or c-Si tip are also observable (Fig. 2a). The intensity of the second-order phonon modes is generally low, perhaps due to the off resonance of the laser frequency with the a-Si direct band gap[39]. Due to the low signal-to-noise ratio, such second-order phonon modes are difficult to be distinguished in standard Raman spectroscopy (Fig. S3b) on a-Si, corroborating the essential role of TERS in this study. Note that the local laser power was set low (25 μW) to avoid laser-induced heating effect on a-Si surface, which is known to be strong in TERS measurements[40]. It has been shown that above a threshold power density value, the laser-induced heating effect could alter the silicon structures and ultimately affect the Raman measurements[41]. The local laser power density used here has shown to keep the a-Si surface intact in our previous report[42]. TERS mapping is able to provide an overview of the intensity distribution of a given vibrational mode. The normalized intensity (at $520\,cm^{-1}$ for TO mode) of each vibrational peak is quantified by the color bar with the correspondence mapping representing the abundance of each chemical moiety. Figure. 2b presents the relative intensity distribution of the a-Si TO mode (approximately centered at $473\,cm^{-1}$). The TERS spectra taken from two spots (denoted as A and B) ~80 nm apart from each other exhibit different TO mode intensity (Fig. 2a). This clearly indicates that the a-Si surface phonon mode unveiled by TERS expresses nanoscale heterogeneity. TERS mapping of the a-Si 2LA mode is shown in Fig. 2c. Clearly, the feature of the intensity distribution of 2LA mode differs from that of the TO mode shown in Fig. 2b, demonstrating the highly heterogeneous feature of the phonon modes on a-Si surface. It is also manifested the spectrum taken from Point C on the TERS map shows a larger 2LA mode intensity than those from Points A and B, but the TO band intensity is lower than the latter two (Fig. 2a). The a-Si 2TO mode at $943\,cm^{-1}$ has a favorable distribution on the upper side of the scanned region (Fig. 2d), different from the distribution of the above-mentioned

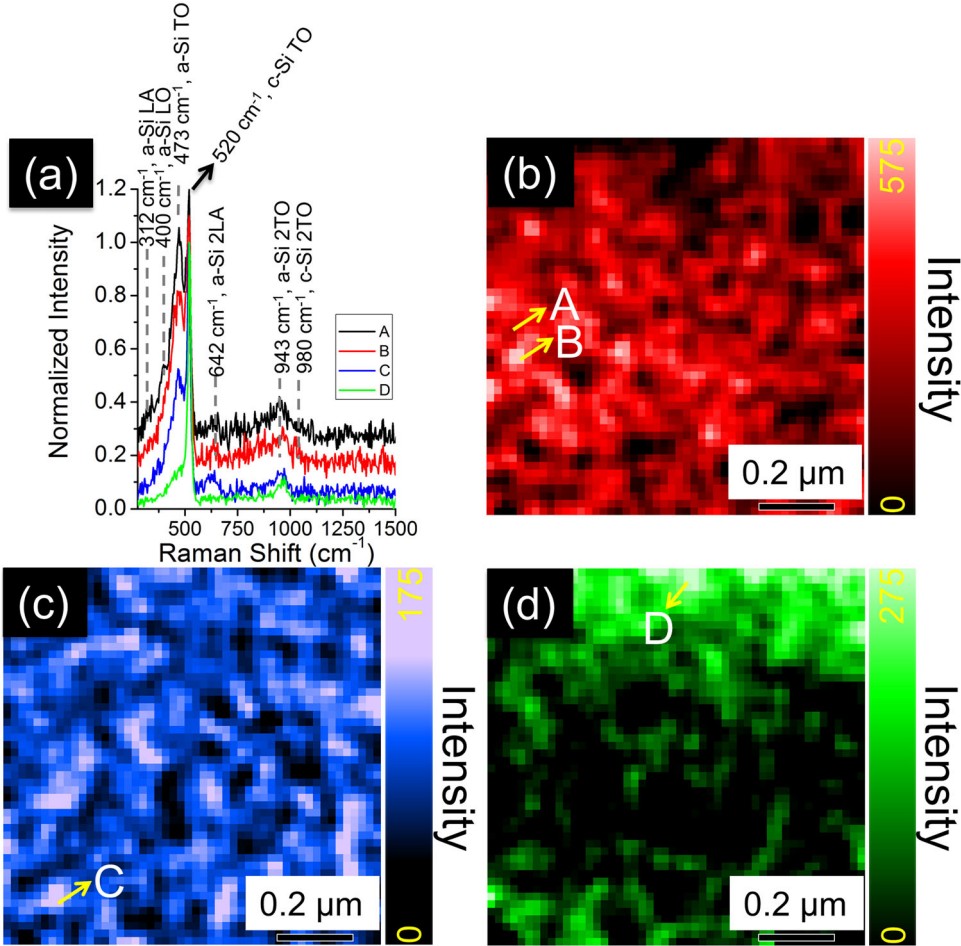

**Fig. 2 TERS spectra from selected locations and TERS mappings based on a single peak. a** The tip-enhanced Raman spectroscopy (TERS) spectra collected from various locations on the amorphous silicon a-Si surface labeled by "A" to "D" in (**b–d**). TERS mapping of an individual peak intensity centered at (**b**) 473 cm$^{-1}$ (a-Si transvers optical or TO mode), (**c**) 642 cm$^{-1}$ (2nd-order longitudinal acoustic or 2LA mode), and (**d**) 943 cm$^{-1}$ (2TO mode). A hybrid TERS mapping combining the peak intensity distribution in (**b–d**) is shown in Fig. S2. The color bar scales the single peak intensity for each TERS mapping.

phonon modes. The TERS mapping generated based on a single-mode intensity clearly demonstrates that (i) TERS is capable of depicting the a-Si phonon mode in nanoscale, and (ii) the abundance of different a-Si phonon modes is highly heterogeneous and varies across the scanned area. The TERS enhancement of the 3rd and 4th order phonon modes is unobvious (See Fig. S5).

The mapping analysis based on the single variant method (i.e. intensity of a single TERS band) is insufficient to lead to a comprehensive understanding of the surface structure of the a-Si[18]. Analysis based on an individual TERS band by nature fails to capture the overall spectral pattern of TERS mapping. To retain the global features embedded in the full set of TERS spectra, we then performed manifold learning and clustering of TERS spectral dataset (see details in method section). Note that the TERS mapping was implemented on a 1 × 1 μm$^2$ area with a 20 nm step size, it yields 2500 TERS spectra in total, with a lateral spectral resolution approximately <60 nm (Fig. S6). Figure 3a is the bootstrapped manifold layout of the set of 2500 TERS spectra, with evolution of manifold layouts during Graph-Bootstrapping iteration procedure shown in Fig. S7. It is worth noting that there is no obvious correlation between the topography as reflected by the AFM height and the TERS mapping (Fig. S8). However, due to the intrinsic technological limitations related to AFM and TERS, the topographic artifacts in the as-measured local TERS intensity cannot be ruled out in the current study, as detailed in the Supporting Information. The 2500 TERS spectra can be

categorized into seven clusters (Fig. 3a) in the 2D manifold space. Here we denote these clusters as "parent clusters" to differentiate from the "child clusters" derived at finer resolution grids in manifold space. The mean TERS spectrum of each cluster is shown in Fig. 3b. An immediate observation on Fig. 3b is that the TO band at ~470 cm$^{-1}$ intensity varies among clusters. The intensity of the second-order phonon peaks also varies, although not much obvious as the TO mode. To quantify the analysis on the TERS bands below 600 cm$^{-1}$, we implemented peak deconvolution of the overlapped phonon vibrational modes of a-Si, including LA, LO, TO modes and c-Si TO mode. A TERS peak deconvolution example is given in Fig. 3c based on the spectrum of cluster 0. It is noteworthy that further improving the dispersion by using grating with more groves per mm didn't assist in better resolving the phonon modes of the a-Si (Fig. S9). The atomistic level local structure of a-Si random network is characterized by the Si–Si bond angle and length distortion ($\Delta\theta$)[37], Si-ring topology (e.g. 5 or 6 membered rings), and voids[43]. The inelastic Raman scattering reflects the full vibrational density-of-states (DOS)[44,45]. Therefore, the local structural change of the a-Si network can be probed by the vibrational frequency change in Raman spectroscopy. Also given that the spring constant for Si–Si bond stretching is much larger than that for bond bending[46,47], bond length distortions are shown to contributes only 1% to the structural gap between a-Si and c-Si comparing with that of the bond angle distortion[48,49]. For TERS measurement, it is only

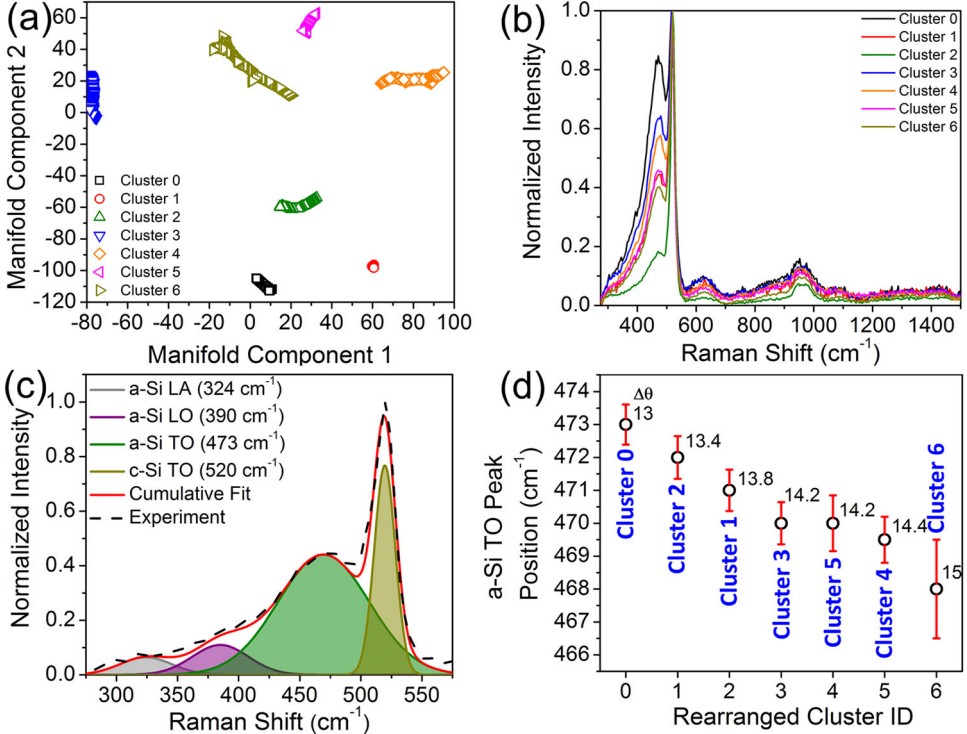

**Fig. 3 The parent manifold layout, mean TERS spectra and transverse optical mode peak position via Graph-Bootstrapping algorithm. a** The parent manifold layout via the Graph-Bootstrapping method (see details in "Method" section) colored by the cluster labels. **b** The mean tip-enhanced Raman spectroscopy (TERS) spectrum for each cluster (spectrum normalized against the c-Si transverse optical or TO mode at 520 cm⁻¹). **c** Peak deconvolution of the silicon TO modes, longitudinal optical (LO) mode and longitudinal acoustic (LA) mode. **d** The TO-band center arranged in declining order. The error bar stands for the Gaussian fitting deviation of the peak center. Δθ is defined as the deviation of the Si–Si bond angle in the a-Si random network from that of the single-crystal Si (109.5°).

sensitive to <10 nm a-Si on the surface (Fig. S1). Another major contribution towards the structural variation of a-Si also comes from the locally under- or over- coordination of the a-Si network[50,51]. However, as noted by an early study, the dangling or floating Si bonds resulted from the under- or over-coordination counts from roughly 1% of the free energy associated with the angle distortion[48]. In this context, we focus on the correlation between the average bond angle distortions (Δθ) and the vibrational frequencies (ω) of the TERS spectra on a-Si. Here the average bond angle distortion is defined the Si–Si angle difference than 109.5°, representing the bond angle in a tetrahedral repeating unit in c-Si.

Using a semi-experimental approach, Vink et al.[38] correlated the TO mode vibrational frequency (in cm⁻¹) to the average bond angle distortions, Δθ (in angular degree), linearly as

$$\omega_{TO} = -2.5\Delta\theta + 505.5. \qquad (1)$$

Vink's computational model was built based on a so-called activation-relaxation technique (ART)[52,53], which yields close-to-experimental a-Si atomic configurations. Equation (1) was found to agree reasonably well with experimental values. Although it is generally accepted that the linear correlation between the feature (i.e. the peak center and width) of the TO Raman peak with Δθ[37,38,45,54,55], there still lacks a designative quantitative agreement among all correlations between the two. Here we do not intend to seek a more accurate correlation between the center of the TO mode and the Δθ, but rather to show that the TO modes measured by TERS can be correlated to the a-Si local structure represented by Δθ. We first rearranged the Raman shift center of the TO mode taken from different clusters in a declining order as shown in Fig. 3d. Using Eq. (1), we can extract the local distortion

angle (marked on Fig. 3d), ranging between 13° and 15°. As noted by Beeman et al.[55], the absolute minimum distortion angle, Δθ was 6.6° for a continuous a-Si network. Here, the local a-Si angular distortion angles taken from different regions are greater than 13°, 12% of the 109.5°. The maximum distortion angle value was found to be 15° for Cluster 6 (Fig. 3d). This corresponds to the a-Si TO mode centered at 468 cm⁻¹, close to the lower limit of the DOS of the optical branch for a-Si[36]. Therefore, the model a-Si thin film used here features a highly disordered random a-Si network on the surface[37]. The highly disordered a-Si structure is also confirmed by the neutron pair distribution function (PDF) shown in Fig. S10, in which a-Si only pertains short-range ordering up to 9.2 Å. Kharintsev et al.[27] reported that the coherent TERS scattering of the amorphous carbon (a-C) could be mapped at nanoscale resolution, given that the hot spot size at the tip apex is smaller than the phonon coherent length of a-C. The a-Si has a phonon coherent length (9.2 Å) approximately one order of magnitude smaller than the TERS hot spot underneath the tip (Fig. S1c). Therefore, the incoherent scattering of a-Si is the major contribution to the TERS spectra in the current study. The calculated strain energy based on the distortion angle is close to the upper limit of that reported by Tsu et al.[37], representing a highly strained a-Si surface (Supporting Information).

So far, we have shown that the surface structure of a-Si explored by TERS can be identified by the MML algorithm without a human-specified physical threshold, which is meaningful to quantify the "poorly-defined" a-Si random network. To solidify this finding, we compare the experimental TERS spectra with the mean TERS spectrum derived from each parent cluster in Fig. 4a. For each cluster, the mean TERS spectrum closely matches with its nearest neighbor, with a mean absolute error

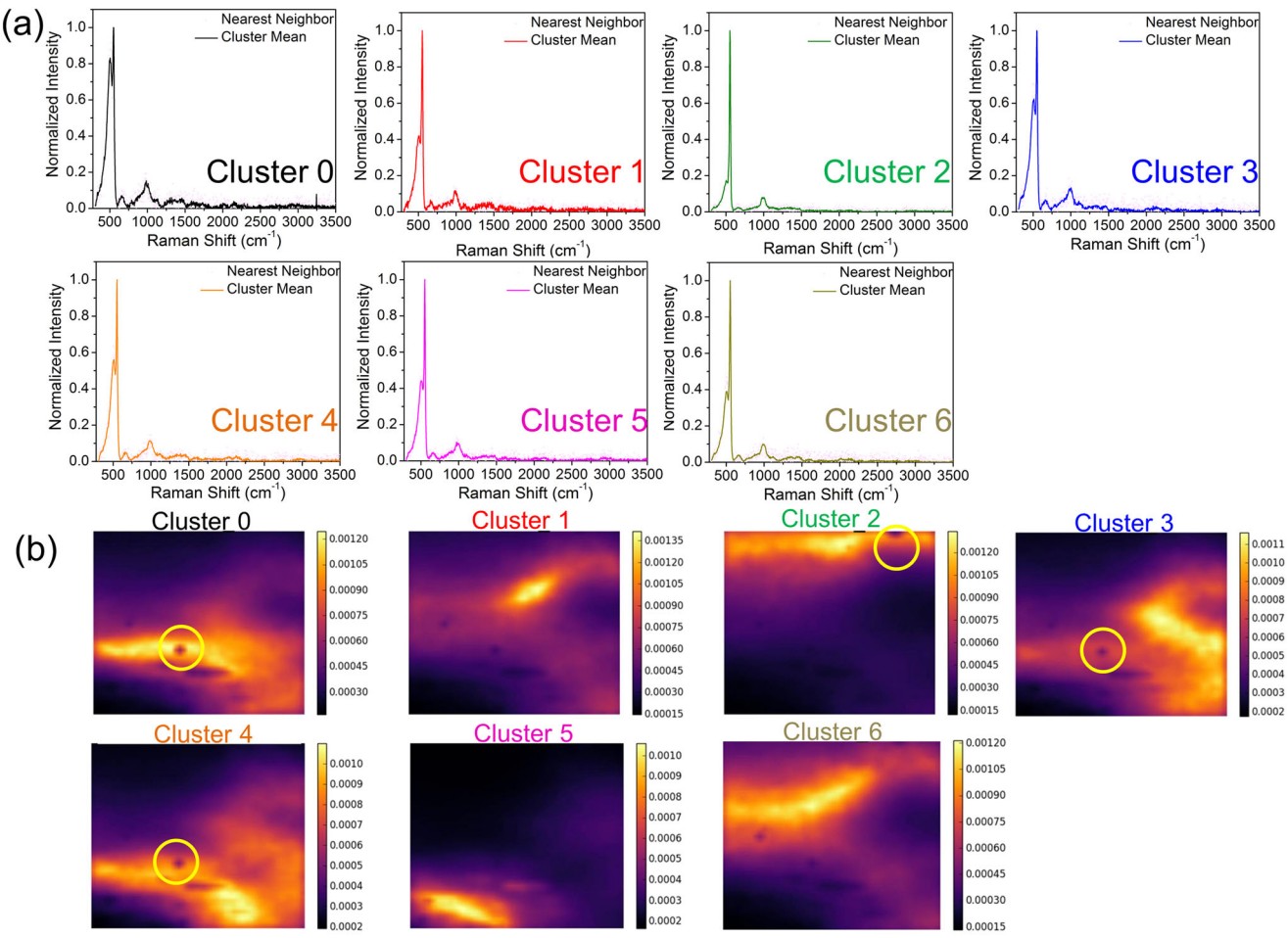

**Fig. 4 Viability evaluation of TERS spectral clustering by the multiresolution manifold learning algorithm and the similarity loading. a** Average tip-enhanced Raman spectroscopy (TERS) spectrum of each cluster and its nearest neighbor in experimental data. **b** Similarity loading of each cluster. The color bar represents the reciprocal of Euclidean distance between the mean TERS spectrum and the as-taken TERS spectrum of each pixel within a given cluster.

percentage <5%. In addition, the pairwise Euclidean distance between the mean TERS spectrum and every as-measured TERS spectrum at each pixel is used to check the variance in each cluster more quantitatively. Specifically, the spatial distribution of variance in the material space can be visualized by the similarity loadings (reciprocal of pairwise Euclidean distances) in Fig. 4b. For the similarity loading of each cluster, a pixel with higher intensity represents a higher similarity between the corresponding experimental TERS spectrum and the mean cluster TERS spectrum.

We note that the black singular patches in the similarity loadings (marked by yellow circles in Fig. 4b) indicate the local TERS spectra different from their surrounding area. To identify these singularities, we performed sub-clustering within each parent cluster by MML. As shown in Fig. 5a, Parent Cluster 2 was divided into 9 child clusters at a finer resolution grid. An immediate observation is that Child Cluster 8 has a distinguished TERS peak centered at 2435 cm$^{-1}$ (Fig. 5b). A closer exploration on the abundance distribution of the TERS spectra (Fig. 5c) indicates that only 7 out of 2500 spectra represented by Child Cluster 9 have the feature of the 2435 cm$^{-1}$ band. This manifests that the characterization strategy developed in this study is sensitive to <0.3% abundance of a structural minority embedded in a large data set. A further inspection on all other singular points shown in the similarity loading maps in Fig. 4 does not reflect TERS spectra carrying any physical meaning. For example, Patent

Cluster 0 was subcategorized into 6 child clusters at the finer resolution grid (Fig. 5d–f), with the Child Cluster 1 exhibiting a sharp spike at 3245 cm$^{-1}$. This spike stems from the cosmic ray commonly seen in Raman spectroscopy. Computationally, the child clustering at a finer resolution grid is adaptive without necessary recalculations of the child manifold layout. However, it necessitates a careful inspection on all stemmed spectra analyzed by the multiresolution method, emphasizing that the domain knowledge of human reasoning is an essential link in Fig. 1.

The assignment of the 2435 cm$^{-1}$ TERS band is not intuitive (denoted as X-mode for now), since no fundamental Si–Si vibrational modes exist in the vicinity of this frequency. It is thus reasonable to assume other surface functional groups present on the a-Si surface. Noting that the a-Si thin film was RF sputtered in an inert (i.e. argon) atmosphere in the current study, the as-sputtered a-Si surface should be enriched in the unbounded Si atoms. Once exposed to air, we expect the components of the air instantaneously react with dangling Si bonds. Further neutron scattering experiment indicates the presence of the protonated spices on a-Si. As shown in Fig. 6a, the first confirmation of the proton presence in a-Si sample is the plot of the structure factor, S(**Q**) (**Q**: scattering vector), with the contributions from both the coherent scattering and incoherent scattering of the sample itself[56]. The anisotropic incoherent scattering cross-section of the proton is 80.26, >37 times of the Si coherent scattering cross-section at 2.163 (inset in Fig. 6a)[57]. Therefore, S(**Q**) plot exhibits a

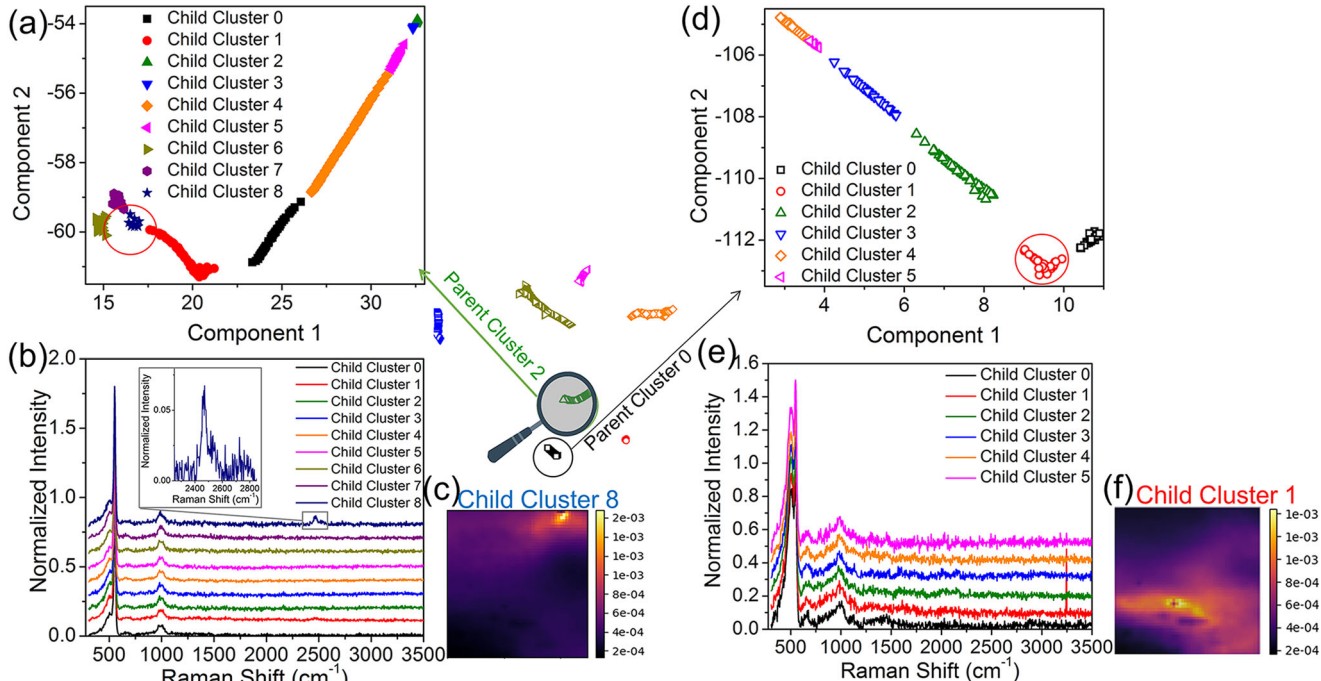

**Fig. 5 The child clusters at the finer resolution grid. a** An overview of child manifold clusters at the finer resolution grid, stemmed from Parent Cluster 2 and (**b**) the corresponding mean tip-enhanced Raman spectroscopy (TERS) spectra of the child clusters. The peak centered at 2435 cm$^{-1}$ of Child Cluster 8 represents unique structural defects different from other child clusters. **c** Similarity loading of Child Cluster 8 that shows a bright blob around the black singular patches in similarity loading of Parent Cluster 2 in Fig. 4. **d** Overview of child clusters stemmed from Parent Cluster 0 and (**e**) the corresponding mean Raman spectra of the child clusters. **f** The similarity loading of Child Cluster 1 that shows a bright blob around the black singular patches in Parent Cluster 0 similarity loading in Fig. 4. The color bar represents the reciprocal of Euclidean distance between the mean TERS spectrum and the as-taken TERS spectrum of each the pixel within a given cluster.

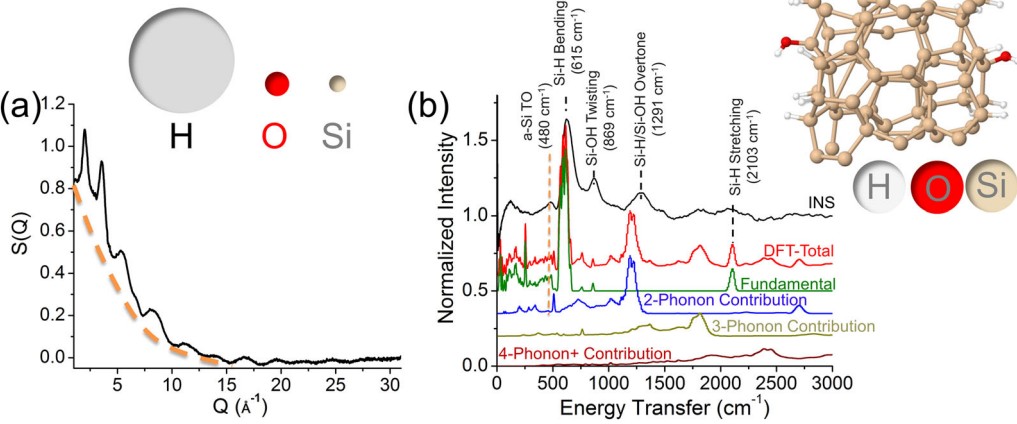

**Fig. 6 The neutron scattering plot and inelastic neutron scattering spectrum of a-Si. a** Neutron scattering plot of the structure factor, S versus the scattering vector, **Q** collected from amorphous silicon a-Si under the same sputtering condition to deposit the a-Si thin film. The dashed curve guides the eye as an indication of the background from incoherent scattering. Inset schematically illustrates the relative size of the incoherent scattering cross-section (XS) of proton (80.260), the coherent XS of oxygen (4.232), and the coherent XS of silicon (2.163). **b** Comparison between the experimental inelastic neutron scattering spectrum of the amorphous silicon a-Si sample and that calculated using DFT based on the atomic conformations shown in the inset image. The calculated total spectrum combines the fundamental vibrational modes and higher order excitations up to 10 orders.

broad background shown in Fig. 6a due to incoherent scattering from H. The reactions between the air water and dangling Si bond results in silane- and siloxane- type moieties on a-Si (see Supporting Information).

The simplest protonated Si compound is Si–H. The reported Raman scattering center for monohydrate (Si-H) ranges between 2030 cm$^{-1}$[58]. and 2090 cm$^{-1}$[59]. Binding more protons to Si leads to blueshift of the Si–H stretching mode, with the maximum frequency found for hydrogenated sputtered Si–H$_4$ compound at

2189 cm$^{-1}$[59]. Given that the surface selection rule complied by TERS differs from those for IR and Raman[18], it is difficult to precisely predict the active TERS vibrational modes for SiH$_x$ compounds in this frequency region.

Inelastic neutron scattering spectroscopy (INS) allows for measuring the vibrational modes in the absence of the selection rules for hydrogen atom[60], thereby exhibiting a comprehensive picture of all possible vibrational modes on the a-Si sample used here. Figure 6b shows an INS spectrum of the a-Si sample. To

unambiguously assign the vibrational modes, we further performed density functional theory (DFT) calculation on a surface hydrogenated a-Si, assuming a "water splitting" mechanism (see supporting information) to form the Si–H and Si–OH functional groups on the a-Si surface (Fig. 6b, inset). The existence of the –H and –OH functional groups in the a-Si sample is validated by the excellent agreement between the experimental INS spectrum and that calculated by DFT in Fig. 6b. The most distinguished proof of the Si–H bond presence is the Si–H bending mode centered at 615 cm$^{-1}$. The presence of the Si–OH groups is evidenced by the Si–OH twisting bands centered at 869 cm$^{-1}$ from INS spectrum[61]. Intriguingly, both experimental and DFT calculated INS spectra show a broad band at around 2430 cm$^{-1}$, assigned to the 4-phonon overtone of the Si–H bending mode. In fact, the Si–H stretching vibrational mode (frequency as $\nu$) is closely related to the electronegativity of the near-neighbor surroundings, with a simplified induction model formulated as[62]

$$\nu = \nu_o + b\Sigma X_A \qquad (2)$$

where $\nu_o$ and $b$ are empirically derived constants and $X_A$ is defined as the stability-ratio (SRX) electronegativity[63]. The values of $\nu_o$ and $b$ were found to be 1741 and 34.7 for molecular compounds, respectively[62]. The sum is over three neighbors, assuming tetracoodination among the neighboring atoms with the central Si. Different atomic species have various values of SRX, namely $X_{Si} = 2.62$, $X_H = 3.55$, and $X_O = 5.21$[61].

In this context, the Si-H stretching mode can further blueshift to above 2250 cm$^{-1}$[61,64]. Based on Eq. (2), the highest possible frequency for Si-H stretching mode calculated is 2283 cm$^{-1}$ for O$_3$–Si–H type compound. However, this value is still 152 cm$^{-1}$ lower than the X-mode shown in the TERS spectrum in Parent Cluster 2, Child Cluster 8 (Fig. 5b). Given that the a-Si thin film is highly disordered on the surface intrinsically, we reason that there possibly exists overly coordinated surface Si atoms of low abundance. We herein define $X_e$ as the excess SRX electronegativity contributed from the excess coordination (>4) to the Si. For the case of X-mode of the TERS band, $X_e$ was calculated at 4.4, close to the reported value for the suboxide silicon as an effective media for a-Si[61]. In this case, the central Si may form the orthosilicate-type compound with the oxygen in surrounding suboxide silicon. Thus, the larger blueshift of the X-mode TERS band than generally reported values originates from overcoordination of the Si–H with the surrounding suboxide (i.e. O$_x$–Si–H$_y$, $x + y > 4$). Since in the same frequency region, as a bulk sensitive spectroscopy technology, INS shows only a broad shoulder (Fig. 6b), the abundance of the overcoordination Si compound should thus be critically low. It is therefore worth emphasizing that the existence of the trace amount (<0.3%) of the surface defects can only be validated by the ultra-surface sensitive technology, TERS, with the structural information distilled by the MML algorithm.

We successfully demonstrate that the nanoscale structural heterogeneity of amorphous Si can be identified and quantified by the synergy between TERS hyperspectral imaging and an unsupervised machine learning-based manifold method. Straightforward clustering and visualization of the manifold structure enable the detection of highly localized conformational changes of a-Si at atomistic level, reflecting the underlying structural and physical essences of the a-Si, including the average Si–Si angle distortions and the strain-free energy, without predefined physical threshold owing to its unsupervised nature. This, in turn, facilitates to set a paradigm to categorize the highly disordered structure for amorphous materials. The multiresolution capability of the MML algorithm allows for mining ultra-low abundance structural information at a finer resolution grid. As a result, a new Raman mode of a-Si surface chemistry embedded in a large TERS dataset can be detected. These insights are valuable for unraveling the nanoscale structure, such as defects of semiconductor devices in both fundamental research and industrial applications.

While the current study solely focuses on a-Si thin film, the combination of ultra-sensitive surface spectroscopy, TERS and the efficient multiresolution manifold learning algorithm should boost scientific discoveries in a broad scope of disciplines, such as solid-state electrolytes, metal–organic framework (MOF), and low-dimension materials, revealing the unknown unknowns to material and domain scientists.

## Methods

**a-Si thin film deposition**. The a-Si thin film was RF magnetron sputtered onto a copper foil using a Si target (99.99%, Kurt J. Lesker) in an in-house sputtering device. Base pressure was below $5 \times 10^{-8}$ Torr and the target to substrate distance was 7 cm. The thickness of the a-Si was 50 nm as measured by a quartz-crystal microbalance (QCM). This type of a-Si thin film was then used for TERS measurement. To increase the amount of material for neutron experiments, 20 μm a-Si films were deposited on the copper substrates. Immediately after the deposition, the a-Si samples were transferred to an Ar-filled glovebox (O$_2$ < 0.1 ppm, H$_2$O < 0.1 ppm) in less than one minute.

**TERS setup and measurements**. A physical vapor deposition (PVD) method was used to fabricate TERS probes from commercial AFM tips (Bruker, OTESPA-R3, resonance frequency = 300 kHz, spring constant = 26 N/m, tip apex diameter = 7 nm)[18]. Briefly, three sequential depositions of chromium (Cr) (2 nm adhesion layer), silver (Ag, plasmonic layer, ~40 nm), and aluminum (Al, protection layer, 1.5 nm) were performed. Al converts to a dense alumina that provides good mechanical and chemical protection without influencing significantly on tip optical properties. TERS tip fabrication details can be found in Reference[65]. A shiny lithium foil was put aside with the sample. There was no observable color change of the lithium foil over the course of TERS measurement, indicating a decent inert gas quality inside the glove box. All TERS measurements were performed on an atomic force microscope (AFM, AIST-NT SMART PROBE) in connection with a Raman spectrometer (HORIBA Co., Xplore) in an argon-filled glove box. For TERS measurements, the 532 nm laser wavelength was chosen with a local power density of 25 μW. The grating number was 600 grooves/mm, and the objective was 100× (N.A. = 0.7). The tapping mode was chosen with oscillation amplitude of 20 nm and a ~2 nm minimum distance from the sample surface for AFM. The mapped area was set at $1 \times 1$ μm$^2$ with a step size of 20 nm. The accumulation time was 0.5 s for each spectral acquisition. Each frame of TERS map represents the intensity (after background correction) of the corresponding vibrational mode that arises from a-Si. The total illumination time for taking a TERS mapping was approximately 20.8 min

**FDTD simulation**. Three-dimensional (3D) finite-difference time-domain (FDTD) simulations (Lumerical Solutions, Inc.) were used to study the electromagnetic (EM-field) distribution between the tip apex and the a-Si sample. The FDTD model is shown in Fig. S1a. Briefly, a silver tip of a 42 nm diameter at the apex was coated with 1.5 nm Al$_2$O$_3$ layer (see Fig. S1b). The tip was set 2 nm from the Si surface. The tip axis had an angle of 10° with the vector of the sample plane. A plane wave of electric field, **E**, polarized along the blue double-arrow in Fig. S1a propagates along the vector. The laser propagation direction is at 55° relative to the perpendicular direction of the Si substrate. The wavelength of the plane wave was set at 532 nm. The spatial mesh size was set at 0.1 nm. The perfectly matched layer (PML) boundary condition (BC) was used for all edges of the simulation box.

**Manifold clustering**. Low-dimensional manifold embedding for TERS measurements is calculated via a modified Graph- Bootstrapping approach[29,30]. Graph-Bootstrapping is an iterative procedure that consists of two main steps: construction of nearest neighbor graph and manifold layout of nearest neighbor graph. During the initialization (iteration 0) of Graph-Bootstrapping, a nearest neighbor graph is calculated based on the high-dimensional TERS measurements, which we call this graph as root graph. Accordingly, we refer the manifold layout of the root graph as root manifold. During the iteration of Graph-Bootstrapping, nearest neighbor graph is reconstructed based on the low-dimensional manifold coordinates of the previous iteration and subsequently manifold layout is updated based on the newly reconstructed graph.

Graph construction and manifold layout follows the way of LargeVis[66]. Approximate nearest neighbor graph construction is calculated via random projection tree[67] and neighbor exploring[68] techniques, given the input dataset **X** = {**X1**, **X2**,…,**Xn**}⊆**R**$^d$. Recall that, during the initialization of Graph-Bootstrapping procedure, the input dataset **X** is the original TERS measurements of high dimensionality. During the iterations of Graph-Bootstrapping, the input dataset **X** is the low-dimensional ($d = 2$) manifold coordinates of the graph of previous iteration). Specifically, the graph is firstly constructed by searching $k$ nearest neighbors via random projection tree method. The graph is then refined via neighbor exploring procedure: (1) Create the max-heap $H_i$ for each node $i$ in the

graph; (2) For each neighbor node $j$ of node $i$, calculate Euclidean distances between node $i$ and each neighbor node $l$ of node $j$, dist$(i,l) = \|x_i - x_j\|$; (3) Push l with dist$(i,l)$ into $H_i$; (4) Pop if $H_i$ has more than $k$ nodes. For each node $i$ and each neighbor node $j$ of $i$, an edge $E(i,j)$ is added to the graph. The weight of symmetric edge $E(i,j)$ is defined in a similar way of t-sne method[69]:

$$w_{ij} = \frac{p_{j|i} + p_{i|j}}{2n}, p_{j|i} = \frac{\exp(-\|x_i - x_j\|^2/2\delta_i^2)}{\sum_{(i,k \in E)} \exp(-\|x_i - x_k\|^2/2\delta_i^2)} \quad (3)$$

To calculate a low-dimensional manifold layout of the graph where each node $i$ of graph is represented by a point $y_i$ in 2D space, a likelihood function is constructed to preserve pair-wise similarities of the nodes in the 2D space[66],

$$L(y_1, y_2, ..., y_n) = \prod_{(i,j) \in E} [f(\|y_i - y_j\|)]^{w_{ij}} \prod_{(i,j) \in \bar{E}} [1 - f(\|y_i - y_j\|)]^{\varepsilon} \quad (4)$$

where f is a probability function set as $f(x) = \frac{1}{1+x^2}$ and $\varepsilon$ is a unified weight. Intuitively, first part of the above equation will keep similar nodes close in 2D space meanwhile second part will tell apart dissimilar nodes in 2D space. The likelihood function can be efficiently maximized with respect to $(y_1, y_2, ..., y_n)$ via negative sampling[70] and alias table sampling[71] of unfolded weighted edges[72], followed by asynchronous stochastic gradient descent[73]. During implementations, we set all hyperparameters default as in LargeVis[66] without any signal pre-processing.

Clustering on manifold is performed by hierarchical density estimates method (HDBSCAN)[74–76], which relies on the mutual reachability distance:

$$D_{mreach,K}(a,b) = \max\{core_k(a), core_k(b), d(a,b)\} \quad (5)$$

where $d(a,b)$ is the original metric distance (Euclidean distance in this paper) between points a and b, $core_k(x)$ is the core-distance of a point x to cover its $k$ nearest neighbors. A minimum spanning tree is firstly built and then condensed upon the hyper-parameter of minimum cluster size, $mc$. The stability of each cluster $C_i$ is defined as:

$$S(C_i) = \sum_{a \in C_i} (\lambda_{max,C_i,a} - \lambda_{min,C_i,a}) \quad (6)$$

where $\lambda$ is the reciprocal of core-distance, $\lambda_{max,C_i,a}$ is the $\lambda$ value at which point $a$ falls out of cluster $C_i$ and $\lambda_{min,C_i,a}$ is the minimum $\lambda$ value at which point $a$ is present in $C_i$. Optimal flat clusters are extracted from walking up the tree to maximize the total stability score over chosen clusters: considering all leaf nodes as initial clusters, if the cluster' stability is greater than the sum of its child, the cluster is selected to be in the current set of optimal flat clustering and all its child are removed from the set. Otherwise, the cluster's stability is set to be the sum of its child stabilities. The main tuning parameter is the minimum cluster size, $mc$. We leave all the other tuning parameters of HDBSCAN as default. For the parent manifold clusters in Fig. 3, we follow a similar procedure in ref. [21] to choose $mc$. We first consider all integer $mc$ values in a wide range of [10,150]. We then fit the trend of total number of estimated clusters against every $mc$ value by the exponential decay function. We choose the $mc$ in the tail region where the total number of clusters tends to be stable. For child manifold clusters at the finer resolution grid as shown in Fig. 5, we set the $mc$ a small value around 0.5% of the total number of TERS measurements.

**Neutron scattering.** All neutron scattering measurements were performed at the Spallation Neutron Source at Oak Ridge National Laboratory (ORNL). Prior to each measurement, the a-Si film was peeled off from the copper substrate in the Ar-filled glovebox ($O_2$ < 0.1 ppm, $H_2O$ < 0.1 ppm) and sealed in a vanadium can (∅ = 6 mm). The total amount of a-Si was 1.368 g and the height was 3.9 cm. An empty vanadium can of the same type was sealed in Ar-glovebox and used as a blank reference.

 a. Neutron total scattering structure function. The structure function data were collected at Nanoscale-Ordered Materials Diffractometer (NOMAD) beamline per a procedure published by Neuefeind et al.[77]. The data collection time was 150 min The structure factor, $S(\mathbf{Q})$ was obtained from a $\mathbf{Q}$ range between 0.5 and 31 Å$^{-1}$.

 b. Inelastic neutron scattering (INS). INS spectra were obtained at the VISION beamline on the same a-Si measured by neutron PDF to assure consistency. The sample was measured in vanadium sample holder at 5 K for about 10 h. The empty sample holder was also measured, and the background spectrum was removed to obtain the spectrum from the sample.

**Molecular Dynamics (MD) simulations.** The models of a-Si were established by conducting molecular dynamics simulations in LAMMPS[78]. Following a melting-and-quench procedure[79], the initial network a-Si was firstly annealed to 2400 K in the $NVT$ ensemble, and then quenched to room temperature (300 K) at a quench rate of $10^{12}$ Ks$^{-1}$. The a-Si models were finally fully relaxed at 300 K in the $NPT$ ensemble. The SW-VBM interatomic potential[80] was used to describe Si–Si interaction. The average coordination number of constructed a-Si models is ~3.99 and quite close to the experimental value (~3.8–3.9)[81–83], validating the high quality of these a-Si models.

**Density functional theory (DFT) calculations.** DFT modeling was performed using the Vienna ab initio Simulation Package (VASP)[84]. The calculation used Projector Augmented Wave (PAW) method [85,86], to describe the effects of core electrons, and Perdew–Burke–Ernzerhof (PBE)[87] implementation of the Generalized Gradient Approximation (GGA) for the exchange-correlation functional. Energy cutoff was 600 eV for the plane-wave basis of the valence electrons. The starting structure of a-Si slab with both surfaces terminated by –H and -OH was generated by MD simulations as discussed above[78]. The thickness of the slab is about 1.5 nm, and the total thickness of the simulation box is 3.5 nm (i.e., 2 nm vacuum). The surface area is about 1.05 nm × 1.05 nm. The simulation box contains 64 Si atoms, and is under 3D periodic boundary condition. The electronic structure was calculated on a 3 × 3 × 1 Monkhorst–Pack mesh. The total energy tolerance for electronic energy minimization was $10^{-5}$ eV, and the maximum interatomic force after relaxation was below 0.01 eV/Å. The vibrational eigen-frequencies and modes were then calculated by the finite displacement method. The OClimax software[88] was used to convert the DFT-calculated phonon results to the simulated INS spectra.

## Data availability
The data that support the findings of this study are available from the corresponding authors upon reasonable request.

## Code availability
The codes that support the findings of this study are available from the corresponding authors upon reasonable request.

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

## Acknowledgements

This work is supported by the U.S. Department of Energy's Vehicle Technologies Office under the Silicon Electrolyte Interface Stabilization (SEISta) Consortium directed by Brian Cunningham and managed by Anthony Burrell. Part of this work was supported by the Center for Nanophase Materials Sciences, a U.S. Department of Energy, Office of Science User Facility at Oak Ridge National Laboratory (S.V.K), Division of Materials Science and Engineering, Biomolecular Materials Program, and Energy Frontier Research Center CSSAS located at University of Washington (X.L.). A portion of this research used resources at the Spallation Neutron Source, a DOE Office of Science User Facility operated by the Oak Ridge National Laboratory. Computing resources for DFT simulations were made available through the VirtuES and the ICE-MAN projects, funded by Laboratory Directed Research and Development program and Compute and Data Environment for Science (CADES) at ORNL. APS acknowledges support from DOE BES Materials Science and Engineering Division for TERS analysis and description. We thank Dr. Dmitry N. Voylov for useful discussion on the TERS experimental setup. We are grateful for fruitful discussions on RF sputtering with Drs. Nancy J. Dudney, Andrew Kercher, and Robert L. Sacci. We thank Drs. Andrew Westover, Jue Liu and Katharine L. Page discussion for NOMAD setup. We thank Michelle S. Everett for strong technical support for NOMAD. This manuscript has been authored by UT-Battelle, LLC under Contract No. DE-AC05-00OR22725 with the U.S. Department of Energy. The United States Government retains and the publisher, by accepting the article for publication, acknowledges that the United States Government retains a non-exclusive, paid-up, irrevocable, world-wide license to publish or reproduce the published form of this manuscript, or allow others to do so, for United States Government purposes. The Department of Energy will provide public access to these results of federally sponsored research in accordance with the DOE Public Access Plan (http://energy.gov/downloads/doe-public-access-plan).

## Author contributions

G.Y. and J.N. conceived and designed the TERS experiments. G.Y. and X.L. conceived the multiresolution manifold learning (MML) for TERS mapping. X.L. developed the codes for MML. G.Y. and X.L. wrote the initial draft. Y.C. and G.Y. performed the INS experiments. Y.C. did INS data analysis and DFT calculations. M.W. performed MD simulations. D.M. performed the neutron PDF experiments and data analysis. A.P.S. assisted in TERS design and analyzed the TERS data. S.V.K. provided supervision on MML design, coding and assisted in proof demonstration. G.V. designed and prepared a-Si thin film samples. J.N. administrated the project and provided the funding source. All authors contributed to writing, revising and reviewing the manuscript.

## Competing interests

The authors declare no competing interests.
