## [Peer Review File · Nature Communications]

REVIEWER COMMENTS

Reviewer #1 (Remarks to the Author):

The paper entitled as “Distilling Nanoscale Heterogeneity of Amorphous Silicon using Tip-enhanced Raman Spectroscopy (TERS) via Multiresolution Manifold Learning” by Guang Yang et. al. tells us about a new approach in TERS methodology, enriched with a multiresolution manifold learning algorithm. AI-assisted TERS allows one better to get insights into the nanoscale structural heterogeneity of disordered materials. Along with the improved capability of TERS, the authors have succeeded, for the first time, experimentally to observe new Raman peaks of c-Si/a-Si species, that have been attributed to highly disordered Ox-Si-Hy modes. This hypothesis has been convincingly proved by inelastic neutron scattering and numerical DFT modelling. Undoubtedly, this study is novel and represents a sound interest in enhanced optical spectromicroscopy. This paper can be recommended for publication in Nature Communications after minor revisions.

1. The authors claims that the power of TERS is straightforward related to the excitation of a plasmon resonance at the tip apex, however, this is not a common case, since there is a lightning rod effect.
2. It is not clear to me that does unraveling vibrational modes with a nanoscale spatial resolution, on page 2, mean? We may speak about the enhancement of intensity of vibrational modes only.
3. On page 4, despite the fact that the light confinement can achieve the extent of 10 nm, we always register a diffraction-limited photon in a distant detector. The spatial resolution in TERS is provided by raster scanning a tip with a nanometer accuracy.
4. Why did the authors use a low power of the incident light of 25 microwatts? Is it related to avoiding photo-induced heating or something else?
5. The authors are intended to capture a Raman map with a spatial resolution less 20 nm, however, a scanning step was chosen to be 20 nm. For reliability, one should have not less 5 points “per resolution”, it means that the scanning step must be ca. 4 nm rather than 20 nm!
6. Because any deconvolution is an ill-posed inverse problem, why didnt the authors use a grating with more grooves per mm (for example, 1800 or echelle one) to physically decompose the overlapped Raman spectrum in Fig. 3 (c) into elementary peaks. Anyway, a high-intensity of the Raman band peaked at 520 cm^{-1} allows to readily do it.
7. It is not obvious how were taken 2500 TERS spectra. One needs to provide more explanations.
8. The authors experimentally didn't demonstrate the TERS enhancement when a tip landed and retracted. Also, a dependence of the TERS intensity vs a sample-tip distance would be welcome.
9. The simulation of the enhancement doesn't take into account the presence of a 2 nm outer Al layer that may shift a dipole plasmon resonance.
10. In the case of a-Si species it would be interesting to recognize contributions from coherent and incoherent enhanced scattering since nanoscale heterogeneity is estimated to be 10 nm, as was earlier shown in papers Phys. Rev. B 2012, 85, 1–8 and J. Phys. Chem. C 2020 (10.1021/acs.jpcc.0c05228) for graphene and a-C, respectively.
11. Why the authors do not show the TERS enhancement of the 3rd and 4th phonon contributions, it would make TERS beneficial in studying of c-Si/a-Si surfaces.

Reviewer #2 (Remarks to the Author):

The authors have reported an interesting method to employ multiresolution manifold to realize multicomponent chemical identification in nanoscale. The manuscript could be published after considering the following comments:

- 1) How to determine the dimension of low-dimensional manifold space and why to reduce the

dimension of hyperspectral TERS data to two-dimensional manifold space?

2) Manifold learning algorithm adopts the idea of local neighborhood when restoring intrinsic invariants, which results in that the stability of the algorithm itself is related to neighborhood selection. How to obtain appropriate neighborhood parameters in the sense of classification?

3) When the noise of data is strong, how stable is the manifold learning algorithm? In other words, under what noise conditions, what algorithm performance can be achieved?

4) The computational complexity of manifold learning algorithm is high, which may hinder its application in practice?

5) In the description of Figure. 1 (line129-131) . "Exploration data analysis such as clustering ... by human reasoning and updating domain knowledge in a loop." The "human reasoning" here is a process of constant trial and error, and then pick out the desired result. In addition, PCA can also perform unsupervised dimensionality reduction, why choose MML?

6) In the description in Figure. 5 (line 285), "child manifold clusters at the finer resolution grid", is the finer network resolution here adaptive or determined after manual experiments? What are the criteria for determination?

7) Typo. In line 147, '...labeled by "A" to "D" in (c-d).' should be replaced by '...labeled by "A" to "D" in (b-d).'

Reviewer #3 (Remarks to the Author):

The manuscript describes the TERS measurement on amorphous Si, and the machine-learning algorithm analysis on the measured phonon peaks. I acknowledge that I do not know enough about the machine learning, and I thus can make comments on only the TERS part:

How do the TERS images correlate to the topography? Have the authors carried out the correlation study on TERS – AFM? I am asking this question because the TERS, as in all forms of NSOM measurements, always has topographic artifacts, which may lead to wrong answers. Specifically, for samples with a finite topographic variation, the tip-particle distance can be changed during the scan even though the tip is feedback-engaged on the surface. The TERS signal intensity rapidly decreases as we increase the tip-surface distance. Thus, the TERS intensity image inevitably contains topographic information, which is unrelated to the materials' property of the particle.

RESPONSE TO REVIEWER COMMENTS

We thank the reviewers for their invaluable comments and suggestions, which we carefully address in this document. The recommendations from all referees are presented in black font, while our itemized responses are shown in blue. The location of each revision is in yellow highlight.

Reviewer #1 (Remarks to the Author):

The paper entitled as “Distilling Nanoscale Heterogeneity of Amorphous Silicon using Tip-enhanced Raman Spectroscopy (TERS) via Multiresolution Manifold Learning” by Guang Yang et. al. tells us about a new approach in TERS methodology, enriched with a multiresolution manifold learning algorithm. AI-assisted TERS allows one better to get insights into the nanoscale structural heterogeneity of disordered materials. Along with the improved capability of TERS, the authors have succeeded, for the first time, experimentally to observe new Raman peaks of c-Si/a-Si species, that have been attributed to highly disordered Ox-Si-Hy modes. This hypothesis has been convincingly proved by inelastic neutron scattering and numerical DFT modelling. Undoubtedly, this study is novel and represents a sound interest in enhanced optical spectromicroscopy. This paper can be recommended for publication in Nature Communications after minor revisions.

We thank the reviewers for the careful evaluation of this manuscript and insightful comments/suggestions to strengthen it. The itemized response is listed below.

1. The authors claims that the power of TERS is straightforward related to the excitation of a plasmon resonance at the tip apex, however, this is not a common case, since there is a lightning rod effect.

We thank reviewer for bringing the lightning rod effect into the scope. While surface plasmon resonance contributes most to the localized enhanced electromagnetic field, the non-resonant enhancement mechanism, such as lightning rod effect also plays an important role to concentrate the electromagnetic field further on the curvature of the TERS tip apex. We made the following revision on Page 2, Line 12.

“TERS is based on strong and local enhancement of the Raman signal by the surface plasmon resonance (SPR) on the metallic tip surface. Additionally, a non-resonant enhanced electromagnetic field (EM-field) occurs at the apex of elongated metallic nanostructure, termed as the lightning rod effect.^{20, 21} The combination of the SPR and the lightning rod effect enables the EM-field to concentrate on the metallized tip apex of a scanning probe microscope (SPM), providing a localized “hot-spot” underneath the SPM tip.”

2. It is not clear to me that does unraveling vibrational modes with a nanoscale spatial resolution, on page 2, mean? We may speak about the enhancement of intensity of vibrational modes only.

We thank reviewer for pointing out this ambiguous statement. We realize that the vibrational modes does not have nanoscale spatial resolution, but rather the surface chemical information carried by the TERS spectra should. This sentence has been revised as follows (Page 2 Line 18)

“Consequently, the Raman scattering signal of the sample within the local hot-spot is largely increased, yielding surface chemical information with a nanoscale lateral resolution.²²”

3. On page 4, despite the fact that the light confinement can achieve the extent of 10 nm, we always register a diffraction-limited photon in a distant detector. The spatial resolution in TERS is provided by raster scanning a tip with a nanometer accuracy.

This is an excellent point. TERS has the capability of probing the nanoscale chemical heterogeneity. Such capability is determined by two factors: a) the nanoscale size of the tip apex where there is maximum field enhancement, and b) the raster step size of the SPM probe on the sample surface.

To elaborate Factor (a), we further performed additional FDTD simulation to illustrate the nanometer size hot-spot underneath the tip apex and added a figure (new Figure S1(d)) to Supporting Information on Page 2.

The following sentence was added in the manuscript on Page 4 Line 8 accordingly.
“The size of a typical hot-spot on the tip apex is on the scale of 10 nm (see Figure S1(d)).”

To better illustrate Factor (b), we added the following sentences on Page 4 Line 9,

“Scattered light enhanced in the hot-spot is registered on a distant detector in the far-field. Using the raster step size of the SPM probe smaller than the hot-spot size enables to record the nanoscale chemical heterogeneity, breaking the light diffraction limit of the standard confocal micro-Raman spectroscopy.^{18, 24}”

4. Why did the authors use a low power of the incident light of 25 microwatts? Is it related to avoiding photo-induced heating or something else?

For the commonly used confocal Raman microscopy, microwatts laser power seems too low to yield reasonable signal-to-noise ratio. And this is exactly what we’ve seen from the Raman spectrum of a-Si thin film when tip was retracted (Figure S4 on Page 4). However, when tip was landed on sample surface, the TERS spectrum had an overall satisfying signal-to-noise ratio. The reviewer is correct, we use low laser power to avoid any possible laser-induced heating effect, and because the enhanced laser light also produce enhanced local heating.

Accordingly, on page 6 Line 26 we added the following paragraph to clarify this point as

“Note that the local laser power was set low (25 μ W) to avoid laser-induced heating effect on a-Si surface, which is known to be strong in TERS measurements.³⁹ It has been shown that above a threshold power density value, the laser-induced heating effect could alter the silicon structures and ultimately affect the Raman measurements.³⁸ The local laser power density used here has shown to keep the a-Si surface intact in our previous report.⁴⁰”

5. The authors are intended to capture a Raman map with a spatial resolution less 20 nm, however, a scanning step was chosen to be 20 nm. For reliability, one should have not less 5 points “per resolution”, it means that the scanning step must be ca. 4 nm rather than 20 nm!

We thank the reviewer for bringing up this critical point. We agree that better imaging could be obtained with a step ~4-5 nm. However, with reducing the step size the mapping time becomes prohibitively long. For instance, if we were to set a 4 nm step size in the current study, it would take 520 min to finish scanning an area of $1 \times 1 \mu\text{m}^2$. Under this circumstance, the thermal drift is unavoidable, which in turn lowers the lateral spatial resolution. Therefore, we had to strike a balance between the step size, acquisition time and scanning range.

To quantify the lateral spatial resolution of the current study, we follow a commonly used method in TERS field – plot the intensity of a vibrational mode across a line cut on TERS mapping and estimate the peak width using Gaussian fit. An additional section was added to the supporting information on Page 6.

“To evaluate the spatial resolution of TERS in our experiments conducted on a-Si, line profiles of the x-mode band intensity along the spots marked by the rectangle in Figure S6(a) is fitted by the Gaussian function (Figure S6(b)). The Gaussian fit has a full width at half maximum (FWHM) of 53 nm, which is slightly larger than the tip diameter (42 nm), and more than twice that of the scanning step size. The method used to evaluate the TERS lateral spatial resolution agrees with other reports.^{3,4}”

Accordingly, we revised the lateral resolution in the manuscript from “20 nm” to “< 60 nm”.

6. Because any deconvolution is an ill-posed inverse problem, why didnt the authors use a grating with more grooves per mm (for example, 1800 or echelle one) to physically decompose the overlapped Raman spectrum in Fig. 3 (c) into elementary peaks. Anyway, a high-intensity of the Raman band peaked at 520 cm^{-1} allows to readily do it.

This is a valid point. A rule of thumb to increase the Raman spectral resolution is to use a higher groove density of the grating. However, no noticeable improvement of the Raman spectral resolution was achieved when using a higher grating of 1800 gr/mm.

We added the following sentence on Page 8 Line 14 in the manuscript –

“It is noteworthy that further improving the dispersion by using grating with more grooves per mm didn’t assist in better resolving the phonon modes of the a-Si (Figure S9).”

Accordingly, the following section was added to the supporting information on Page 8

“The use of high-resolution grating can usually improve the spectral resolution. We further explored the possibility to physically deconvolute the overlapped Raman modes of the a-Si by using a grating with 1800 gr/mm. To avoid the interference of the 520 cm^{-1} peak from c-Si, only a-Si thin film was focused on by a standard confocal Raman microscopy. However, there was no noticeable improvement of the Raman spectral resolution compared to that collected by a 600 gr/mm grating, as seen from Figure S9.”

7. It is not obvious how were taken 2500 TERS spectra. One needs to provide more explanations.

We thank the reviewer for this helpful comment. We added the following sentence on Page 7 Line 9

“Note that the TERS mapping was implemented on a $1 \times 1 \mu\text{m}^2$ area with a 20 nm step size. It yields 2500 TERS spectra in total.”

8. The authors experimentally didn't demonstrate the TERS enhancement when a tip landed and retracted. Also, a dependence of the TERS intensity vs a sample-tip distance would be welcome.

To address this comment, we included a “tip-in and tip-out” test to compare the Raman spectra collected with and without enhancement. The enhancement-factor was also estimated based on this test. An additional figure was added on Page 4 in the Supporting Information. The following paragraph was added on the same page in SI:

“It was found that even when the tip was retracted by 5 nm from the a-Si surface, the 1st order TO mode intensity at $\sim 473 \text{ cm}^{-1}$ of a-Si is significantly reduced (Figure S4), demonstrating that the near-field in the vicinity of the tip contributes most to the TERS signal. The near-field intensity (I_{near}) of the TO mode is estimated to be 850. With tip retracted from the a-Si surface by 5 nm, the intensity of the TO mode reduced to approximately 100 (i.e. far field intensity, $I_{far}=100$). The laser spot size is estimated to be $1 \mu\text{m}$ with the penetration depth of roughly 20 nm. The far-field sample volume (V_f) is thus estimated at $0.016 \mu\text{m}^3$. The illuminating spot of the near-field underneath the tip as shown in Figure S1 has an estimated radius of 5 nm. Assuming the same penetration depth in a-Si as the far-field, the near-field scattering volume (V_n) is then estimated to be $1.57 \times 10^{-6} \mu\text{m}^3$. The enhancement factor, EF can then be calculated as,¹²

$$EF = \frac{I_{near}}{I_{far}} \cdot \frac{V_{far}}{V_{near}}$$

The calculated EF in this study is $\sim 8 \times 10^4$.”

9. The simulation of the enhancement doesn't take into account the presence of a 2 nm outer Al layer that may shift a dipole plasmon resonance.

We emphasize that the tip has non-metallic Al_2O_3 layer. To address the reviewer comment, a new FDTD model with Al_2O_3 thin layer covered on the tip apex was built. A scheme of such a model and the new set of the simulation results were added on Page 2 Figure S1 in supporting information. It is noteworthy that with a thin Al_2O_3 layer coating, the simulated enhancement factor is reduced by estimated 2 orders of magnitude, likely because of the increased distance between the metallic tip and the sample surface. We included the following paragraph to discuss this point on the same page in SI:

“Three-dimensional (3D) finite difference time domain (FDTD) simulation (Lumerical Solutions, Inc.) was used to better study the locally enhanced EM field on the Si surface. The FDTD model was shown in Figure S1(a). Briefly, a silver tip of a 42 nm diameter at the apex was coated with 1.5 nm Al_2O_3 layer (see Figure S1(b)). The tip was set 2 nm from Si surface. The tip axis had an angle of 10° with the vector of the sample plane. A plane wave of electric field, \mathbf{E} , polarized along the blue double-arrow in Figure S1(a) propagates along the vector. The laser propagation direction is at 55° relative to the perpendicular direction of the Si substrate. The wavelength of the plane wave was set at 532 nm. The spatial mesh size was set at 0.1 nm. The perfectly matched layer (PML) boundary condition (BC) was used for all edges of the simulation box (resort to the reference¹ in supporting information for mathematical details).”

10. In the case of a-Si species it would be interesting to recognize contributions from coherent and incoherent enhanced scattering since nanoscale heterogeneity is estimated to be 10 nm, as was earlier shown in papers Phys. Rev. B 2012, 85, 1–8 and J. Phys. Chem. C 2020 (10.1021/acs.jpcc.0c05228) for graphene and a-C, respectively.

This is a very interesting point that we didn't consider in the initial manuscript. After a careful evaluation, we conclude that the incoherent scattering is the major contribution to the TERS spectra. This conclusion can be reasoned as follows.

According to Reference J. Phys. Chem. C 2020 (10.1021/acs.jpcc.0c05228), the coherent TERS response can be observed only if the excitation spot size is comparable to or smaller than the phonon coherence length. The hot spot in the current study is estimated to be 10 nm in diameter, one order of magnitude larger than the coherence length of the a-Si thin film evaluated by neutron PDF.

Accordingly, we added the following paragraph to the current manuscript on Page 9 Line 9

“The highly disordered a-Si structure is also confirmed by the neutron pair distribution function (PDF) shown in Figure S10, in which a-Si only pertains short-range ordering up to 9.2 Å. Kharintsev et al.²⁶ reported that the coherent TERS scattering of the amorphous carbon (a-C) could be mapped at nanoscale resolution, given that the hot spot size at the tip apex is smaller than the phonon coherent length of a-C. The a-Si has a phonon coherent length (9.2 Å) approximately one order of magnitude smaller than the TERS hot spot underneath the tip (Figure S1(c)). Therefore, the incoherent scattering of a-Si is the major contribution to the TERS spectra in the current study.”

The plot of neutron PDF used to estimate the coherence length of the a-Si thin film was added to Page 9 in the supporting information, with the following detailed discussion on the same page

“Neutron PDF plot was used to estimate the phonon coherence length of the a-Si thin film in the current study. As shown in Figure S10, in contrast to crystalline silicon, a-Si does not possess long-range translational order, as manifested by the fact that its pair distribution function (G(r)) does not show noticeable peak at above 9.2 Å. G(r) correlates to the neutron powder diffraction data through the Fourier transform of the scattering structure function S(Q) by

$$G(r) = \frac{2}{\pi} \int_0^{\infty} Q[S(Q) - 1] \sin(Qr) dQ$$

The TERS lateral light confinement width, L_w is estimated by $\sim \sqrt{2hr_o}$,⁹ in which h is the tip-sample distance, h = 2 nm and r_o is the curvature radius of the tip apex and estimated to be 21 nm). L_w is thus estimated to be 9.17 nm, agreeing fairly well with that estimated from FDTD simulation as shown in Figure S1. Note that the coherent TERS response can be observed only if the excitation spot size is comparable to or smaller than the phonon coherence length,⁹ we thus reason that the incoherent scattering is the major contribution to the TERS spectra.”

11. Why the authors do not show the TERS enhancement of the 3rd and 4th phonon contributions, it would make TERS beneficial in studying of c-Si/a-Si surfaces.

This a fair point. After going over the TERS spectra within the scanned area, we concluded that the enhancement of the 3rd and 4th order phonon contributions by TERS is not obvious, at least under the current experimental conditions (new Figure S5) was added on Page 5 in the supporting information.

Most probably this is due to insufficient signal-to-noise ratio to unveil the 3rd and 4th order phonon modes in our measurements.

To better illustrate this part, we added the following paragraph on **Page 5** of the SI,

“To explore the possible TERS enhancement of the 3rd and 4th order phonon modes of the a-Si, we went through all TERS spectra in the scanned area. Using 473 cm⁻¹ as the peak center for 1TO mode for a-Si, there is no noticeable TERS peaks showing up at around ~1419 cm⁻¹ (3TO) and 1892 cm⁻¹ (4TO) shown in Figure S5. To more quantitatively illustrate this point, we resorted to the DFT calculation shown in Figure 6. Assuming that TERS has the same relative intensity of the fundamental 1TO mode versus higher phonon modes with that for INS, the intensity of the 3TO is 15.4% of 1TO, and that of the 4TO mode is 6.1%. The average signal-to-noise ratio is estimated to be 15%. Therefore, the reason that the 3TO and 4TO phonon modes are lacking is probably due to the unfavorable signal-to-noise ratio under the current experimental conditions. ”

Reviewer #2 (Remarks to the Author):

The authors have reported an interesting method to employ multiresolution manifold to realize multicomponent chemical identification in nanoscale. The manuscript could be published after considering the following comments:

We sincerely thank the reviewer for the detailed examination of this work. The comments have helped drastically improve the illustration of the manifold learning algorithm in the manuscript.

1) How to determine the dimension of low-dimensional manifold space and why to reduce the dimension of hyperspectral TERS data to two-dimensional manifold space?

This is a valid point. The choice of dimension of low-dimensional manifold space serves the following purposes: a. to reduce the computational burden for clustering (regardless the specific clustering algorithm used); b. to facilitate straightforward visualization of clusters and human-machine interaction for scientific discoveries, which is also detailed further in response to reviewer’s comment 5). We chose to use two-dimensional manifold space, which better serves the two above mentioned purposes. In addition, two-dimensional manifold space is a popular choice in the manifold learning literature.

2) Manifold learning algorithm adopts the idea of local neighborhood when restoring intrinsic invariants, which results in that the stability of the algorithm itself is related to neighborhood selection. How to obtain appropriate neighborhood parameters in the sense of classification?

This comment is important for understanding the algorithm implementation.

Specifically, the key hyper-parameter is the number of the nearest neighbor, k . As detailed in the Method part, Graph-Bootstrapping is an iterative procedure built on the recent state-of-art LargeVis [Ref 64] algorithm. We follow the default rules used in LargeVis to set tuning parameters. The default value of k is 150. One advantage of LargeVis claimed by authors was: “the hyper-parameters of LargeVis are also much more stable over different data sets”.

However, from our earlier work (Refs [28,29]), we found that naïve implementation of LargeVis for experimental spectroscopy datasets resulted in the bulk module, making it difficult for clustering algorithm to further explore local details inside the bulk module. Specific examples can be found in Figures 4a and c in Ref[28], Figure 2b in Ref[29] and Figure S7 in this work (imagine the layout without color labels). Note that, the number of the nearest neighbor; k is a positive integer with maximum limit to be the total number of collected spectroscopy curves, N . For scanning probe microscope (SPM) type datasets, N is usually on the scale of thousands. Efforts of exploring effects of different k are thus time-consuming. To our best knowledge, there is no mathematical rule to determine the “optimal” k universally. In the current study, we tried k around 50, 150 and 500, all resulting in a bulk manifold module. This is also consistent with the stability claim by LargeVis authors.

To explore intrinsic structures and present them in a clearer way, we proposed the Graph-bootstrapping procedure in earlier work (Ref[28,29]) that consisting of a. iteratively reconstructing the graph based on the manifold coordinates and b. subsequently recalculating the manifold layout positions based on the reconstructed graph, following the same principle probability model used in LargeVis.

Here comes to a new intriguing question open to future exploration: instead of tuning the number of nearest neighbor k , how many iterations does one need in the Graph-Bootstrapping procedure?

In our earlier work (Ref[28,29]), we found 1 iteration was enough. However, we found 4 iterations is a feasible choice in this work. One may refer to Supplementary Figure S7 for the manifold layouts during iterations of graph-bootstrapping procedure. Figure R1 below is the manifold layout during 5th iteration of graph-bootstrapping. We see the number of groups is overwhelming for reasonable data analysis. Accordingly, we do clustering and sub-clustering tasks based on 4th-order manifold layout in a multi-resolution way.

Figure R1: manifold layout during 5th iteration of graph-bootstrapping

Indeed, comparing 5th-order manifold and 4th-order manifold involves “human-reasoning” as mentioned in comment 5). Mathematically, we cannot give a closed-form rule to determine the “optimal” number of iterations. As detailed in response to comment 5), we pose that for scientific discoveries, one may not solely rely on the mathematical rules to arrive at “optimum”. Nonetheless, we believe the TERS spectroscopy datasets provide a favorable playground to bridge machine learning, physics and material science communities.

3) When the noise of data is strong, how stable is the manifold learning algorithm? In other words, under what noise conditions, what algorithm performance can be achieved?

This is an excellent question. For the TERS dataset collected over the amorphous-silicon shown in Figure 4a), we can see this dataset has more noise at the high frequency range. This is partially due the ultra-low laser power used to take the TERS imaging as detailed in the response to Comment 4 from the 1st Reviewer. Nonetheless, manifold learning and clustering can distill the local Si-Si bond angle distortion statistically and pinpoint miniature defect structures out of thousands of TERS spectra as demonstrated in Figures 4 and 5. However, this set of dataset is unfavorable for quantifying the stability of the manifold learning algorithm versus the noise level.

To verify the stability of manifold learning with respect to noise level, we collected a Raman map on a semi-crystalline poly(ethylene oxide) (PEO) thin film. The PEO Raman map was scanned over a 100 x 100 μm region, with a total number of 14400 spectra collected. The laser power was set at 1 mW, 40 times that used for TERS mapping collected from a-Si, resulting in a promoted signal-to-noise ratio of the each Raman spectrum. The PEO has two phases at room temperature - crystalline phase and amorphous phase. With this prior knowledge, we expect to see two clusters of the all Raman spectra. Then we artificially impose white noise ϵ_i , at different levels to individual PEO Raman spectrum and see how it affects the manifold learning results.

T_i denotes the spectral intensity at the frequency w_i . We consider three different noise levels, $A = 0.05, 0.15, 0.3$, respectively.

$$\epsilon_i \sim A * T_i * N(0,1)$$

Note that for $A=0.3$, the white noise counts for as high as 30% of the Raman intensity.

Figure R2 below illustrates the robustness of manifold learning over noise.

Manifold layout of Raman spectra without white noise ($A=0$)

Manifold layout of Raman spectra with white noise

Without the artificial white noise, all 14400 Raman spectra of the PEO can be categorized into two clusters, representing the crystalline phase and the amorphous phase. For the noise cases, we color-coded the manifolds using the same set of labels as that from the noise-free dataset. For $A = 0.05$ (meaning the white noise can be up to 5% of Raman intensity), manifold layouts clearly display two groups. When the noise level increased to $A=0.15$, we can see a tiny portion of the amorphous cluster manifold overlaps with the crystalline counterpart. Interestingly, at noise level $A = 0.3$ (meaning the white noise is 30% of Raman intensity), the manifold layout shrank to a single module. However, points of the identical color still resemble a cluster.

4) The computational complexity of manifold learning algorithm is high, which may hinder its application in practice?

We agree that the complexity of the manifold learning algorithm increased the computational cost. However, we shall point out that thanks to rapid progress of the algorithm development in machine learning community, recently developed state-of-the-art manifold learning algorithms (LargeVis and variants) significantly reduce the computational complexity to $O(N)$, making it linear to number of observations. For example, for spectroscopy datasets corresponding to a 120 X 120 pixel size image with each spectroscopy of length 800, the data dimension is 14400 by 800, resulting in a text file of ~150 MB. The whole computational procedure takes less than 15 minutes on a personal laptop (Intel i7-8750H CPU@2.2GHz, 16GB RAM).

5) In the description of Figure. 1 (line129-131) . "Exploration data analysis such as clustering ... by human reasoning and updating domain knowledge in a loop." The "human reasoning" here is

a process of constant trial and error, and then pick out the desired result. In addition, PCA can also perform unsupervised dimensionality reduction, why choose MML?

We thank reviewer for bringing these excellent points. To begin with, from methodological point of view, we would like to refer to the following paper

—*Clustering: Science or Art?*¹ published at ICML that is written by Professors Von Luxburg U, Williamson RC, and Guyon I:

“If clustering researchers want real impact in applications, then it is time to step back from a purely mathematical and algorithmic point of view. What is missing is not “better” clustering algorithms but a problem-centric perspective in order to devise meaningful evaluation procedures.”

Based on the a-Si TERS dataset in this work, we reason why MML is preferred over the PCA for such data analysis followed by reasoning why “a purely mathematical and algorithm point of view” is not enough.

PCA Analysis:

To utilize PCA, one must first determine the number of principal components (PCs). One rule of thumb to determine PC number is the Scree plot as shown in Figure R3. X axis is the number of PCs and Y axis is the variance percentage explained by the specific PCA component. The number of PCs is selected around the turning point, which is 6 in this case. We can see the explained variance drops to almost zero at the 6th PC.

Figure R4 displays the 6 PCA component “spectra”. The positive peaks in PCA component are not located (although closely) at the physically correct frequency index. The existence of the negative TERS peaks cannot be reasoned by practice, as there are no negative TERS peaks in the real world. It ultimately leads to the question of how important the human reasoning is in the current study.

Figure R3: Scree plot used to select the number of PCs in PCA.

¹ Von Luxburg, Ulrike, Robert C. Williamson, and Isabelle Guyon. "Clustering: Science or art?." Proceedings of ICML Workshop on Unsupervised and Transfer Learning. 2012.

Figure R4: All component spectra in PCA

Necessity of Human Reasoning:

The human reasoning is indispensable for scientific discoveries. Mathematically, there is nothing wrong for the above PCA analysis. However, it is fairly difficult for spectroscopists to extract meaningful information from the resultant component spectra. The attempt to gain the categorized a-Si TO mode fails, as evidenced by the negative TO mode in PC-2, 3, 5 and 6.

From a domain application perspective, we are treating each TERS spectrum as the “digital signature” or a “fingerprint” of a certain amorphous material structure. Given that the prior knowledge of the material structure is lacking for the 2500 TERS spectra of the a-Si (i.e. those spectra are unlabeled), spectroscopists need to deal with two basic tasks:

- Can 2500 TERS spectra within the scanned area be represented by characteristic TERS spectra?
- If so, how are these representative TERS spectra distributed?

No matter how “advanced” a machine learning algorithm is, above tasks have to be verified manually in the end. Because for all scientific exploration tasks, the new knowledge and discoveries have to be digested and reasoned by the human beings instead of the machine itself. Machine learning serves as the tool to facilitate and accelerate above tasks for human reasoning.

Coming back to mathematical point of view, we now further illustrate why manifold learning is more suitable for above exploration tasks. Manifold learning begins with building the nearest neighbor graph based on pair-wise similarity (Euclidean distance) of TERS curves. In this context, manifold learning by nature targets on extracting relationships among TERS curves. Then the nearest neighbor graph is preserved and visualized in low-dimensional manifold layout via a principled probability model, such that aggregating points in manifold space imply high similarities between the corresponding TERS curves, facilitating visualization and clustering.

Yet, independent of specific machine learning algorithm, we still have to verify the results physically to answer above tasks:

- a. Can 2500 TERS spectra within the scanned area be represented by characteristic TERS spectra? – We need to inspect mean TERS spectrum for each cluster and see if the mean TERS spectrum indeed represents the experimental ones. In Figure 4(a), we can see for every cluster, the mean TERS spectrum closely match with its nearest neighbor, with mean absolute error percentage <5%.
- b. If so, how are these characteristic TERS spectra distributed? — We can visualize the distribution quantitatively via the similarity loadings shown in Figure 4(b). In particular, similarity (inverse of Euclidean space) between mean TERS spectrum and as-measured TERS spectrum is calculated. Therefore, in Figure 4(b), a pixel with high intensity represents a high similarity between the corresponding experimental TERS spectrum and the mean cluster TERS spectrum.

A reasonable example can be seen from the black singular patches in Figure 4, which are indicative of material “defects” to Raman spectroscopists. It further drives us to visualize and sub-clustering the parental manifold clusters at the finer resolution grid shown in Figure 5. It turns out that two child-clusters correspond to a new TERS band structure and the commonly seen cosmic ray spike. Without “human reasoning”, the sub-clustering would not have been implemented and this new TERS band might not have been discovered. Therefore, we would like to reemphasize the domain knowledge and human resonating is an essential link in Figure 1.

Methodologically, above analysis falls under the general suggestions of “Exploratory — confirmatory”, “Unsupervised — supervised” in the section 4, “A suggestion for future research” in the article “*Clustering: Science or Art?*”. Based on the current work and our earlier work in Refs [24, 25], we would like to move one step further by asking “Clustering meeting Physics: Science with Art?”.

6) In the description in Figure. 5 (line 285), “child manifold clusters at the finer resolution grid”, is the finer network resolution here adaptive or determined after manual experiments? What are the criteria for determination?

This is a great point. Computationally speaking, child clustering at finer network resolution is adaptive. The reason is that it does not require additional computational efforts to recalculate the child manifold layout. By nature, a child clutter is a more enriched sub-cluster from the parent cluster (Figure 3a) in the manifold space. Child cluster has the same coordinates of each manifold point as the parent cluster. This is the multi-resolution property of manifold learning we would like to emphasize. To our best knowledge, this should be the first demonstration of multi-resolution property of manifold learning this way.

Regarding the criteria for determination of the child clustering, as detailed in response to comment 5), one reason to stimulate us to do the sub-clustering in Figure 5 is the existence of a few singular points in the similarity loading (Figure 4). Similarity loading check is suggested in general. Strictly speaking, even without Figure 4, we may still perform the sub-clustering for each parent cluster and then find the miniature sub-clusters corresponding to the hidden material structures.

In an analogy to the “Google map”, the manifold learning encodes the high-dimensional information into a “2D map”. We demonstrate herein the enriched details in manifold layout can be explored in a multi-resolution way. It is an application specific and user decision to whether “zoom in” or not.

Accordingly, we added the following sentence on **Page 11 Line 10** in the manuscript

“Computationally, the child clustering at finer resolution grid is adaptive without necessary recalculations of the child manifold layout.”

7) Typo. In line 147, ‘...labeled by “A” to “D” in (c-d).’ should be replaced by ‘...labeled by “A” to “D” in (b-d).’

We thank the reviewer for the careful proofread. The typo was corrected in Figure 2 caption.

Reviewer #3 (Remarks to the Author):

The manuscript describes the TERS measurement on amorphous Si, and the machine-learning algorithm analysis on the measured phonon peaks. I acknowledge that I do not know enough about the machine learning, and I thus can make comments on only the TERS part:

We appreciate the reviewer’s professional comments on the technical part of TERS. Since Reviewer #2 focuses on machine learning part, further comments on TERS part will definitely help further improve the quality of the current manuscript.

How do the TERS images correlate to the topography? Have the authors carried out the correlation study on TERS – AFM? I am asking this question because the TERS, as in all forms of NSOM measurements, always has topographic artifacts, which may lead to wrong answers. Specifically, for samples with a finite topographic variation, the tip-particle distance can be changed during the scan even though the tip is feedback-engaged on the surface. The TERS signal intensity rapidly decreases as we increase the tip-surface distance.

Thus, the TERS intensity image inevitably contains topographic information, which is unrelated to the materials’ property of the particle.

This is an excellent point. We have carefully inspected the TERS mapping and the topography. There seems to be unobvious correlation between the two. However, we note that the topological artifacts are inevitable due to the intrinsic technological limits as the reviewer properly mentioned. Therefore, the following sentence has been added to the manuscript on **Page 8, Line 2** as

“However, due to the intrinsic technological limitations related to AFM and TERS, the topographic artifacts in the as-measured local TERS intensity cannot be ruled out in the current study, as detailed in the Supporting Information.”

We should also point out that the topographical artifacts influence the intensity of the a-Si phonon mode. The manifold clustering used in this study recognizes the overall TERS pattern. For example, the analysis of the local distortion angle of the Si-Si in Figure 3(d) is based on the TO mode shift instead of the absolute intensity. Therefore, the topographical artifacts do not alter the major conclusions in the current study.

We have included a detailed discussion in the supporting information on **Page 7**:

“Upon analysis on several spots on the TERS mapping and the corresponding AFM height image, we failed to find a direct correlation between the a-Si surface TERS and the topography. Shown in Figure S8 (a), the black singular point presenting marked by the circle has a relative height of -99 nm as shown in Figure S8(c) marked by Point a. Point b along the Arrow ab in Figure S8(c) has an equal relative height of Point a. As discussed in Figure 5, Point a presents the X-mode in TERS spectrum, whereas Point b does not. Another example is shown in Figure S8(b), where the ellipse marks Cluster 3 TERS spectra, in which Point c has the same height of Point d in Figures S8(c) and (e). However, the TERS spectrum taken from Point d falls into Cluster 6 (Figure 4). Therefore, it is manifest that TERS spectra taken from the sampling points of the same height in the AFM topography do not necessarily bear the same similarity.

At this stage, we cannot rule out the topographic artifacts that affect the TERS mapping due to the intrinsic technological limitations related to AFM and TERS. AFM images are taken due to the physical interaction between the scanning tip and the sample surface through piezoelectric ceramic scanners (definition). Any factors affecting this interaction influence the resultant AFM images. These factors include ai) geometrical shape and size of the tip; aii) hysteretic behavior of the piezoelectric scanner; aiii) thermal drift of the sample etc. ⁴ Regarding Factor ai, the tip has an end radius of ~20 nm in our current system. Therefore, any feature smaller than 20 nm would be convoluted on lateral adjacent points. However, the FDTD simulation (Figure S1) clearly shows that the size of the hot spot underneath the tip is on the scale of 10 nm, thereby providing a TERS map with the lateral resolution surpassing the resolution of the AFM image. Factor aii originates from the fact that that the same driving signal of the piezoelectric scanner does not correspond to the same position when scanning back and forth. It leads to a slight lateral shift of a step-like feature on AFM image. Factor aiii is less significant in the current study as we lowered the Raman laser power to 25 μ W and minimized the laser irradiation time to mitigate the laser induced thermal effect. In addition to the intrinsic AFM topological artifacts, surface roughness could affect the TERS mapping due to the following reasons, bi) tip-sample distance varies during the tip raster across the sample plane to affect the TERS intensity of the a-Si; bii) nanoscale roughness may modulate the TERS signal intensity by up to 10 folds. ⁵ All above-mentioned factors lead to the mismatch of the TERS mapping (i.e. intensity of a vibrational mode) and the sample topography.

However, we point out here that the topographical artifact does not alter the major conclusion in the current study. The clustering algorithm applied is not dependent on the intensity of one or two TERS bands, but rather on the overall similarity of the TERS spectra within the scanned area. This is evident from Figure 3(d), in which the a-Si TO mode varies in the peak center, which is related to the local Si-Si bond and length distortions. The topographical artifact influencing the TERS peak shift of amorphous silicon is a subject worth exploring in future. To accommodate such a study, one will need well-controlled geometric factors for both the tip and the sample. A better model system should include a sharper TERS tip, and a smoother a-Si thin film coated on an atomic smooth substrate with reduced piezoelectric scanner hysteresis.”

REVIEWERS' COMMENTS

Reviewer #1 (Remarks to the Author):

Since the authors gave exhaustive answers for all questions and considerably improved the text of the manuscript as a whole, I would recommend to publish it as it is.

Reviewer #2 (Remarks to the Author):

The authors have addressed my questions with satisfactory. The manuscript can be accepted for publication in its present form.

Reviewer #3 (Remarks to the Author):

The revised manuscript fully and correctly addresses the concern I have raised, and I thus recommend the manuscript to be published to Nat. Comm. without further revision.